# Brain-Behavior-Immune Interaction: Serum Cytokines and Growth Factors in Patients with Eating Disorders at Extremes of the Body Mass Index (BMI) Spectrum

**DOI:** 10.3390/nu11091995

**Published:** 2019-08-23

**Authors:** Mariarita Caroleo, Elvira Anna Carbone, Marta Greco, Domenica Maria Corigliano, Biagio Arcidiacono, Gilda Fazia, Marianna Rania, Matteo Aloi, Luca Gallelli, Cristina Segura-Garcia, Daniela Patrizia Foti, Antonio Brunetti

**Affiliations:** 1Department of Health Sciences, University Magna Graecia of Catanzaro, 88100 Catanzaro, Italy; 2Department of Medical and Surgical Sciences, University Magna Graecia of Catanzaro, 88100 Catanzaro, Italy

**Keywords:** eating disorders, anorexia nervosa, binge eating disorder, obesity, cytokines, growth factors, inflammation

## Abstract

Alterations of the immune system are known in eating disorders (EDs), however the importance of cytokine balance in this context has not been clarified. We compared cytokines and growth factors at opposite ends of BMI ranges, in 90 patients classified in relation to BMI, depressive and EDs comorbidities. Serum concentrations of interleukin (IL)-1α, IL-1β, IL-2, IL-4, IL-6, IL-8, IL-10, interferon-gamma (IFN-γ), tumor necrosis factor-alpha (TNF-α), monocyte chemoattractant protein-1 (MCP-1), vascular endothelial growth factor (VEGF), and epidermal growth factor (EGF) were determined by a biochip analyzer (Randox Labs). Differences were calculated through ANOVA. Possible predictors of higher cytokine levels were evaluated through regression analysis. IL-1α, IL-10, EGF, and IFN-γ were altered individuals with anorexia nervosa (AN) and binge eating disorder (BED). Night-eating was associated with IL-8 and EGF levels, IL-10 concentrations with post-dinner eating and negatively with sweet-eating, long fasting with higher IFN-γ levels. IL-2 increase was not linked to EDs, but to the interaction of depression and BMI. Altogether, for the first time, IL-1α, IL-10, EGF, and IFN-γ were shown to differ between AN and HCs, and between AN and individuals with obesity with or without BED. Only IL-2 was influenced by depression. Dysfunctional eating behaviors predicted abnormal concentrations of IL-10, EGF, IL-8 and IFN-γ.

## 1. Introduction

Eating disorders (EDs), such as anorexia nervosa (AN), bulimia nervosa (BN) and binge-eating disorder (BED) are characterized by a complex psychopathology and dysfunctional eating behavior [1] that could influence the neuroendocrine and immune systems [2]. As a consequence, alterations in immunological functions have generated great interest according to the hypothesis that pro- and anti-inflammatory cytokines may have reciprocal influences on appetite and food intake, and promote a cascade of biochemical events culminating in the deregulation of neurohormones, neuropeptides and neurotransmitters, which are responsible for the development and self-perpetuating of EDs and eating-related maladaptive behaviors [3,4]. However, the importance of abnormal cytokine production for the possible subsequent onset of these illnesses has not been clarified yet. The most recent reviews on the relationship of cytokines with EDs have produced differing conclusions, suggesting a complex role of these mediators [5,6].

Food intake, eating behaviors and nutritional status (both malnutrition and chronic hypercaloric nutrition) are very influential factors on the immune response and cytokine production [7,8].

Deregulation of the cytokine network observed in EDs could reflect a primary disorder or could be related to their complications, including impaired nutrition, psychopathological and neuroendocrine factors [9]. In particular, elevated Tumor Necrosis Factor-alpha (TNF-α) and interleukin (IL)-6 plasma levels and lower levels of IL-2 and Tumor Growth Factor-beta 2 (TGF-2) in patients with AN compared to healthy controls (HCs) [5,6,10], probably linked to impaired nutrition and weight loss, ref. [10] have been described.

On the other hand, obesity is characterized by a chronic low-grade inflammation, in which a moderate increase in circulating levels of pro-inflammatory cytokines and a macrophage infiltration of adipose tissue (able to secrete various biologically active adipocytokines) are evident [11]. Previous studies have documented partial links between excess weight gain or body fat mass and increased levels of circulating cytokines, such as IL-6 and TNF-α [12,13]. Only in one study IL-6 levels also correlated positively with body mass index (BMI) [14]. IL-8, a pro-inflammatory chemokine produced by several cell types, including macrophages, was shown to be elevated in adults with obesity [15]. Although lower serum levels of IL-10 were described in obesity [16], elevated levels of IL-10 have been reported in many inflammatory conditions [17], indicating potential attempts to inhibit inflammation by downregulating pro-inflammatory cytokines [18].

In light of the immunological abnormalities found in patients with EDs, the role of cytokines in the onset and progression of EDs requires further investigation [19]. Changes in inflammatory mediators, in specific brain regions (induced by dieting and food-associated stressors), can contribute to the onset of eating-related maladaptive behavior that could play a role in the interplay between hypothalamic-pituitary- adrenal (HPA) axis activity and immune regulation [20].

Moreover, chronic stress, mood disorders comorbidity, fasting or binging, weight variability, the severity and nature of hypocaloric or hypercaloric diet, should be more attentively considered when studying cytokine levels and their production in EDs patients [21].

Some data about higher inflammatory indexes related to BED-obese patients have been already described [22], but to the best of our knowledge, no studies regarding cytokine levels among these patients have been published [19]. Comparing plasma cytokine levels in AN and BED represents a unique opportunity to test the relationship between inflammatory markers and specific eating disorders, in which abnormal BMIs and specific EDs coexist. Our hypothesis is that circulating levels of cytokines could be influenced by specific eating behaviors and psychiatric comorbidity, among normal weight, underweight and individuals with obesity. Accordingly, we aimed to compare cytokines and growth factors levels, searching differences between patients with and without EDs and depression. We also aimed to elucidate differences among BED-obese patients according to their eating dysfunctional behaviors. This is the first study that measured serum levels of pro-inflammatory interleukins IL-1-α, IL-1β, IL-2, IL-4, IL-6, IL-8, interferon-γ (IFN-γ),TNF-α, monocyte chemoattractant protein 1 (MCP-1), the anti-inflammatory interleukin IL-10 and growth factors, such as vascular endothelial growth factor (VEGF) and epidermal growth factor (EGF) in individuals with depressive and EDs comorbidity and dysfunctional eating behaviors at opposite ends of the BMI range.

## 2. Materials and Methods

### 2.1. Patients and Study Design

We designed a transversal, observational study in the routine management of consecutive outpatients seeking a treatment for EDs. All patients were recruited at the University Unit of psychiatry from September 2017 to December 2018.

Individuals were selected according to the following eligibility criteria: aged 18–65 years; drug-naïve; diagnosis of AN restrictive type or BED according to DSM-5, patients with obesity without comorbid EDs, normal-weight healthy controls without any psychiatric diagnosis according to DSM-5; ability to answer a self-reporting questionnaire; understanding the process in which they were involved.

The exclusion criteria were: individuals aged under 18 or over 65 years; patients with AN purging type or BN according to DSM-5; normal-weight persons with any psychiatric comorbidity according to DSM-5; patients affected by diabetes mellitus, neurological or other medical conditions that might affect cognitive functioning; hormonal and pharmacological treatment with the potential to induce metabolic changes or cognitive impairment; smokers; pregnancy or childbirth over the previous 12 months. Written informed consent was obtained from all participants and the study was approved by local Ethics Committee (Comitato Etico Regione Calabria, sezione Area Centro: 565/D.G. 01.08.2017).

### 2.2. Measures and Procedures

Physiological features, pharmacological treatments, current and past history of mental and physical disorders were assessed. Participants’ height and weight were measured using a portable stadiometer (Seca 220, Seca GmbH & Co., Hamburg, Germany) and a balance scale (Seca 761), wearing light indoor clothing and no shoes, with standing height measured to the nearest 0.1 cm and body weight to the nearest 0.1 kg at 8.00 a.m.; then, their BMI (kg/m^2^) was calculated. Participants’ BMI were classified according to the WHO guidelines [23].

#### 2.2.1. Eating Psychopathology

An experienced investigator initially conducted an in-depth assessment of participants’ abnormal eating behaviors (namely grazing, emotional eating, craving for carbohydrates, sweet eating, post-dinner eating, night eating, binge eating, hyperphagia, social eating, long fasting, skipping meals, reducing portions) during the previous 6 months. Behaviors were considered to be present when all the related items were answered “yes” and if the behavior had caused clinically significant impairment or distress.

#### 2.2.2. Psychopathology Measures

A trained psychiatrist subsequently administered the Structured Clinical Interview for the DSM-5 (SCID-5-CV) [24] to assess psychiatric disorders and the Eating Disorder Examination (EDE 17.0 D) [25] to assess ED disorders. Then, patients were required to fill the 21-item self-report Beck Depression Inventory (BDI-II) [26]. Scores between 0–9, 10–16, 17–29 and ≥30 indicate minimum, mild, moderate and severe depression, respectively. Cronbach’s alpha in the present research was 770. BDI score was used as a covariate measure in statistical analysis.

#### 2.2.3. Serum Cytokines and Growth Factors Determination

Venous blood was collected after 12–14 h fasting. After centrifugation of blood samples at 3500 rpm for 10 min, sera were aliquoted and stored in non-absorbing polypropylene tubes and frozen at −80 °C until use. The serum concentrations of 12 different cytokines and growth factors (IL1-α, IL-1β, IL-2, IL-4, IL-6, IL-8, IL-10, IFN-γ, TNF-α, MCP-1, VEGF, and EGF) were simultaneously determined using the biochip analyzer Evidence Investigator (Randox Labs, Crumlin, UK) and the “Citokine Array I” kit (Randox) according to the manufacturer’s instructions, and previous descriptions [27]. Briefly, the biochip array technology is based on a multiplex chemiluminescent immunoassay testing, in which specific antibodies, bound to the biochip surface, allow the immobilization of specific analytes. Subsequent bonds of the immunocomplexes with secondary, specific antibodies labelled with horseradish peroxidase trigger a luminol-based chemiluminescent signal emission. The light signal generated from each of the test regions is then detected using an imaging technology and quantified with a stored calibration curve. Relative quality controls were used to monitor accuracy and precision at three different levels of cytokine concentrations. Analytical intra-run and inter-run imprecision errors (CV) were both <10% for each biomolecule tested.

### 2.3. Statistics

Data are presented as means, standard deviations (SD), frequencies and percentages (%). Factorial ANOVA was conducted to compare the main effects of diagnosis and BMI categories and the interaction between diagnosis and BMI on the cytokine levels controlling for BDI and sex as fixed factors and age as covariate. Diagnosis included four levels (healthy control, anorexia nervosa, binge eating disorder, obesity) and BMI consisted of five levels (underweight, normal weight, obesity class I, II, III). A series of forward stepwise linear regression analysis were run considering dysfunctional eating behaviors as potential predictors of serum concentration of cytokines and growth factors. Statistical Package for Social Sciences version 21 (SPSS, Chicago, IL, USA) was used to perform the statistical analysis.

## 3. Results

### 3.1. Study Population and Cytokine Levels

Overall, 90 out of 101 participants (89.1%) met the inclusion criteria and agreed to participate in the study. Table 1 shows the main socio-demographic and clinical characteristics of the cohort.

The total sample consisted of 28 males (31%) and 62 females (69%) with an average age of 36.7 ± 13.2 (18–65) years. Four subgroups were analyzed according to their clinical characteristics: anorexia nervosa, obesity with or without binge-eating disorder, healthy controls. According to the WHO guidelines, 15 participants resulted to be underweight (BMI < 18.5 kg/m^2^), *n* = 21 normal weight (BMI 18.5–24.9) and *n* = 20 class I obesity (BMI 30–35 kg/m^2^), *n* = 9 class II obesity (BMI 36–39.9 kg/m^2^) and *n* = 25 class III obesity (BMI > 40 kg/m^2^). For statistical purposes, patients with obesity were split into two groups: Obesity 1, corresponding to class I (BMI 30–35 kg/m^2^; *n* = 20), and Obesity 2, corresponding to class II / III (BMI > 35 kg/m^2^; *n* = 34).

Table 2 illustrates the means, standard deviations and quartiles of serum cytokines in relation to BMI.

#### 3.1.1. Interleukin-1α

The main effect linked to diagnosis of EDs yielded an F ratio of F(3.53) = 2.936, *p* < 0.05, η^2^ = 0.145 indicating a significant difference between healthy controls (1.4 ± 0.3), anorexia nervosa (0.7 ± 0.3) binge eating disorder (0.3 ± 0.2) and individuals with obesity (0.4 ± 0.2). There was a not significant effect of BMI (F(3.53) = 0.671; *p* > 0.05) and depression (F(1.53) = 1.456; *p* > 0.05) on IL-1α, as well as a not significant interaction effect either.

#### 3.1.2. Interleukin-2

The main effect linked to diagnosis of EDs yielded an F ratio of F(3.53) = 1.404. *p* >0.05; the main effect of BMI yielded a F ratio of F(3.53) = 1.580. *p* > 0.05 and the main effect of depression yielded an F ratio of F(1.53) = 1.818; *p* > 0.05 thus indicating that the effect of diagnosis (healthy controls = 4.3 ± 2.5; anorexia nervosa = 12.7 ± 2.4; binge eating disorder = 3.1 ± 1.9; obesity = 2.6 ± 1.9). BMI (underweight = 3.9 ± 2.5; normal-weight = 10.3 ± 2.3; obesity-1/class I obesity = 4.1 ± 2.3; obesity-2/class II/III obesity = 2.4 ± 1.4) and depression were not significant for IL-2. Nevertheless, the interaction effect of diagnosis for depression (F(2.53) = 5.899; *p* < 0.01; η^2^ = 0.185) and the interaction effect of BMI for depression (F(2.53) = 4.189. *p* < 0.05; η^2^ = 0.139) were significant. Taken together, these results suggest that the interactions of diagnosis and BMI for depression have an effect on IL-2 levels.

#### 3.1.3. Interleukin-10

All effects were statistically significant at the <0.01 significance level. The main effect for EDs diagnosis yielded an F ratio of F(3.53) = 4.762; *p* = 0.005, η^2^ = 0.216 indicating a significant difference between healthy controls (1.2 ± 1.3), anorexia nervosa (5.9 ± 1.2), binge eating disorder (1.9 ± 0.9) and individuals with obesity (0.8 ± 0.9). The main effect for BMI yielded an F ratio of F(3.53) = 4.799, *p* = 0.005. η^2^ = 0.217, indicating a significant difference between underweight (1.0 ± 1.3), normal-weight (4.8 ± 1.1), obesity 1/class I obesity (2.1 ± 1.1) and obesity 2/class II/III obesity (1.1 ± 0.7). The main effect of depression yielded an F ratio of F(1.53) = 1.147, *p* > 0.05. There was no significant interaction effect.

#### 3.1.4. Interferon-γ

All the effects were statistically significant at the <.05 significance level. The main effect for EDs diagnosis yielded an F ratio of F(3.53) = 6.090; *p* = 0.001; η^2^ = 0.260, indicating a significant difference between healthy controls (1.9 ± 0.9), anorexia nervosa (4.9 ± 0.9), binge eating disorder (1.3 ± 0.7) and patients with obesity (1.1 ± 0.6). The main effect for BMI yielded an F ratio of F(3.53) = 7.591, *p* < 0.001, η^2^ = 0.305, indicating a significant difference between underweight (0.4 ± 0.9), normal-weight (4.9 ± 0.8), obesity 1/class I obesity (1.2 ± 0.8) and obesity 2/class II/III obesity (1.6 ± 0.5). The main effect of depression yielded an F ratio of F(1.53) = 2.825, *p* > 0.05.There was no significant interaction effect.

### 3.2. Regression Analysis

To check whether specific dysfunctional eating behaviors can be considered potential predictors of serum concentration of cytokines and growth factor, regression analysis was carried out (Table 3).

Night eating significantly influenced the serum concentrations of IL-8 (F(8.212), *p* = 0.005) and EGF(F(7.603), *p* = 0.007); “post-dinner eating” and “not having sweet eating” were significantly associated with higher levels of IL-10 (F(4.259), *p* = 0.007). Finally, “long fasting" was significantly associated with higher levels of IFN-γ (F(4.067), *p* = 0.047).

## 4. Discussion

The balance between immune modulatory mediators at the extremes of the BMI spectrum on the one hand, and the coexistence of EDs or depression on the other hand, is an important element in the comprehension of the complex process of the immune response to malnutrition. A deregulated cytokine pattern is evident in this cross-sectional study.

We measured a wide range of circulating markers involved in the inflammatory process in individuals according to depressive symptoms, EDs comorbidity and dysfunctional eating behaviors. We excluded patients with BN because the levels of cytokines could be influenced by other medical comorbidities linked to frequent long term purging behaviors, self-induced vomiting and laxatives or diuretics use, or prolonged periods of starvation that could influence inflammatory response and immune system, causing chronic inflammation of oral and digestive system and impairment of liver and kidney function [28,29].

Consistent with our hypothesis, certain inflammatory markers, as IL-2, IL-10, IFN-γ and IL1-α, were found to differ between AN and HCs, as well as between AN and participants with obesity with or without BED.

Our exploratory analyses also identified potential confounding variables of these markers, including BMI and depression. Some dysfunctional eating behaviors (frequent among persons with obesity) predicted concentrations of IL-10, EGF, IL-8 and IFN-γ.

Despite previous findings had already described elevated levels of pro-inflammatory cytokines in AN compared to healthy individuals including IL-6 [6,30], TNF-α [31,32,33,34], IL-1 [31,32,33,34], we did not find significant differences in IL-1β, IL-4, IL-6, IL-8, TNF-α among groups regarding either ED diagnosis or BMI. Other authors also failed to find any differences in serum levels of IL-1, IL-6 or TNF-α [3,35,36,37,38] in both AN or BN patients compared to HCs.

Many reasons have been considered to explain these contradictory results. Deregulation of cytokines in AN could reflect a primary disorder or might result from starvation [4]. The severity of underweightness, the type of hypocaloric diet (fat, and/or protein-calorie deficient diet) chronic stressors or comorbid affective disorders have been thought to interfere with this deregulation [4]. Moreover, AN may be associated with enhanced activity of the pro-inflammatory cytokines without concomitantly increased serum levels of these cytokines [6].

Marcos et al. [39] noted that an anorexic diet, although deficient in carbohydrates and fat, is relatively sufficient in protein [40], which could explain the maintenance of an intact functional immune system compared to the case of protein energy malnutrition. Protein deficiency reduces the ability of monocytes to produce cytokines [41]. Thus, AN may be distinguished from other conditions of nutritional deprivation, which are associated with severe immunosuppression [42]. It was therefore suggested that plasma from AN patients has sufficient nutrients to sustain a normal lymphocyte transformation response [43,44]. Furthermore, according to others, cytokine levels in these patients could be also influenced by the length of illness and malnutrition that turns into normality after re-feeding [32].

We observed that IL-1α, but not IL-1β serum levels were significantly influenced by EDs diagnosis, being essentially higher in AN patients or patients with obesity with or without BED than in HCs, while they were not influenced by BMI or depression. These results are in line with previous studies [9]. Moreover, it has been observed that IL-1 α and several other cytokines (e.g., TNF-α, IL-6, and IFN-γ) could maintain food intake suppression in animals and humans [45]. Therefore, it was speculated that patients with AN might also have an increased production of this cytokine, that could be involved in the self-maintenance of illness [7].

Raymond et al. [7] have also found that IL-1α production by stimulated peripheral blood mononuclear cells (PBMC) was higher in patients with obesity than in the control group. We did not find significant differences in the groups with obesity, with and without BED, and we are unaware of previous studies of IL-1α in BED.

Since complex interactions occur between cytokines and the central nervous system, differences in the capacity of patients to evoke a compensatory mechanism through either the neuroendocrine system or the autonomic nervous system, explain the results obtained. In fact, highly significant inter-individual variations in the production of cytokines by PBMC have been also reported among healthy individuals [46].

IL-2 and IFN-γ, the major type-1 cytokines produced by T helper-1 (TH-1) cells, are known to be increased under conditions of chronic stress and suppressed by glucocorticoids [47]. It has been reported that in vitro IL-2 production and serum levels are lower in patients with AN than in healthy controls [10,32,35], reflecting a frequently depressed TH-1-like activity in AN, which may be not specific to the pathophysiology of EDs, but explained by stress conditions in AN [35].

Conversely, three studies identified no differences in IL-2 production between AN and HCs [32,48,49], while increased IL-2 and IL-2 receptors (sIL-2r) serum levels were found among patients with depression [50].

Our results seem to resume and confirm these findings: we found that serum levels of IL-2 were not significantly influenced by ED diagnosis or BMI, but by the interaction effect of depression and BMI. Therefore, the presence of a comorbidity with depression among AN patients seems to explain these patterns. In the current study, we observed that serum levels of IFN-γ were influenced by ED diagnosis. We found raised levels of IFN-γ in AN patients, compared to HCs and BED with obesity and not BED individuals, with no influence from depression. A possible explanation of these results could be that fasting, a condition that frequently occurs among AN patients, might contribute to an increased production of IFN-γ [7]. Indeed, most AN patients in this work were newly diagnosed, so that the average length of illness was lower than a year and patients not severely malnourished. Thus IFN-γ values could be related to the short duration of malnutrition.

Previous findings related to IFN-γ production in AN are controversial [8,47,48,50,51]. Differences could be explained by a significant correlation between the duration of the illness and the production of IFN-γ [48]. In AN, lower levels could be related to a functional defect of peripheral lymphocytes [51], while re-feeding could be associated with the recovery of cytokine production [52]. Coherently, the capacity to produce IFN-γ partially improves with weight gain and restoration of the menstruation cycle [51]. Despite previous findings [12,13,14], we did not find any significant differences regarding IL-6 and TNF-a serum levels in our sample, probably because of the small sample size.

We also found that IL-10 levels were significantly influenced by EDs diagnosis (being essentially higher in AN and lower in BED), by BMI (being higher in normal-weight and lower in class II/III obesity but not influenced by depression. To the best of our knowledge, this is the first time that serum concentrations of IL-10 have been compared in AN and BED patients and were found to be altered in both of them. IL-10 synthesized by several cell types within multiple organs including the spleen, is a potent anti-inflammatory cytokine [53]. Lower IL-10 production capacity has been demonstrated in obesity [16], implicating that the lack of IL-10 may, at least in part, contribute to the prevalent pro-inflammatory profile expressed in obesity [54,55].

We found lower IL-10 levels in individuals with obesity (with and without BED) compared to AN and HCs, confirming previous findings [55]. On the other hand, elevated IL-10 levels have been reported in many inflammatory conditions, including malnutrition [18], and may indicate attempts to inhibit inflammation by downregulating the secretion of pro-inflammatory cytokines [18].

Few previous studies have evaluated IL-10 in AN patients [9]. We found higher levels of IL-10 in AN patients compared to individuals with obesity and HCs. We hypothesize that our result, not yet clarified by previous studies, may be a consequence of malnutrition.

In fact, the overproduction of IL-10 and the downregulation of pro-inflammatory cytokines have often been associated with immunosuppression [54,55,56,57,58]. IL-10 is a particularly important anti-inflammatory mediator that regulates both innate and adaptive immune responses [57,59,60,61] with a severe reduction in spleen cellularity [59]. Previous studies found different patterns of immune markers variability among patients with obesity, during progressive weight loss [27,61].

Finally, we evaluated possible predictors of higher cytokine levels. The association with IFN-γ and long term fasting have been underlined before by us, and confirmed by a previous study [7]. We hypothesize that elevated levels of serum IL-10 in not habitual sweet eaters, could be inversely proportional to metabolic and weight consequences of this eating behavior.

The most original results are related to EGF. Growth factors are a group of polypeptides that modify cell proliferation, constituting a distinct subgroup in endocrinology. According to regression analysis, only EGF seemed to be linked to eating dysfunctional behaviors in our sample. EGF is a member of the family of EGF-like molecules [62]. Early studies demonstrated its stimulatory effect on epidermal proliferation and inhibitory effect on gastric acid secretion [63]. EGF was initially purified by Cohen from salivary gland extracts [63,64], and a previous study showed that EGF secretion capacity varied greatly according to the rate of saliva secretion [65]. The secretory activities of salivary glands are highly individual and change throughout the day according to diurnal rhythm, meal intake and the health status of the individual. The amount of salivary secreted proteins could influence serum levels [66]. Daraf et al., [67] studied the diurnal effects of EGF and IL-8 and the impact of meal intake on secretory output. It is well recognized that eating exerts a powerful stimulus upon salivary glands and because of long-acting reflex mechanisms, proteins that are packed in the secretory granules are released following a meal [66].

Since EGF is produced and stored in the salivary glands and secreted reflexively [68], we could speculate that serum levels of EGF in patients with night eating behaviors, could be influenced by salivary and gastric acid nocturnal secretion. Future evaluation of EGF salivary secretion should be performed to better clarify these results.

### Strengths and Limitations

This is the first study to measure several inflammatory markers and growth factors in patients with AN, individuals with obesity with and without BED, compared to HCs, according to dysfunctional eating behaviors. We also assessed how depressive comorbidity could influence results in our participants. Few previous studies have included such variables [19], which, in the current study, have allowed us to explore the relationship between the type and severity of illness and inflammatory markers. In addition, regarding HCs, the median values of a number of the assessed markers are similar to those observed in previous studies, suggesting that our HC group is a valid comparison group. Several limitations should be noted. First, the sample is small, which limits the power in this study and due to its exploratory nature, no corrections for multiple testing have been made, thus increasing the likelihood of significant results. Additionally, it must be mentioned that the AN group was younger (mean age = 25.1 years) than the HC (mean age = 31.8 years) and patients with obesity (mean age = 41years) groups. The limited power in this study precludes more complex analyses, such as incorporating several confounding variables or considering the effect of nominal confounders on inflammatory markers Furthermore, we have to specify that our sample was acquired from treatment-seeking individuals, so this population may differ from previous studies and this might effect to results.

Overall, future investigations on inflammatory markers in AN and BED need to ensure that such confounders are assessed and reported, and if possible, accounted for in statistical analyses.

Limitations of cross-sectional studies preclude conclusions about causality. Future longitudinal studies could be useful to clarify variability of cytokine levels according to other variables. For example, if inflammation is particularly pronounced during the early illness phase, anti-inflammatory agents could be tested, especially in early phase patients, and randomized controlled trials should be considered. More research is needed to better understand the pathophysiologic relationship and trajectory of inflammation in AN and BED during worsening and improvement of BMI status.

## 5. Conclusions

This exploratory study compared a broad range of inflammatory markers and growth factors, both in AN and in BED patients, many of which had not been previously assessed. IL-1α, IL-10, EGF, and IFN-γ, for the first time, were shown to be altered in individuals with AN and BED in comparison to HCs. We also considered age, BMI, depressive comorbidity, as potential confounding variables of concentrations of the inflammatory markers. Only serum levels of IL-2 were influenced by depression. Our findings suggest that future research should include covariates in analyses of this relationship to explore whether this may account for some of the group differences in inflammatory markers observed in the current study. Finally, given that these inflammatory markers function as part of a complex network, future studies in larger samples, should consider developing a composite score of cytokine concentrations.

## Figures and Tables

**Table 1 nutrients-11-01995-t001:** Demographic and clinical characteristics of the study participants.

		Healthy Control (HC)		Anorexia Nervosa (AN)		Binge Eating Disorder (BED)		Patients with Obesity	
		*n* = 21		*n* = 14		*n* = 27		*n* = 28	
		Mean	SD	Mean	SD	Mean	SD	Mean	SD
Age		31.8	12.3	25.1	11.6	41.0	11.8	41.9	11.4
BMI		21.3	2.7	16.6	1.1	38.4	7.9	42.2	10.5
BMI categories ^§^	Underweight	3	14.3	12	85.7	0	0	0	0
	Normal weight	18	85.7	2	14.3	1	3.7	0	0
	Obesity 1	0	0	0	0	8	29.6	10	35.7
	Obesity 2	0	0	0	0	18	66.7	18	64.3
Beck Depression Inventory		10.0	10.6	24.8	15.0	25.0	9.7	10.7	9.5
Eating behaviors (yes) ^§^	Long fasting	4	19	10	71	2	7	6	21
	Hyperphagia	1	5	0	0	10	37	20	70
	Binge	1	5	1	7	26	96	5	18
	Grazing	0	0	2	14	25	92	8	30
	Emotional eating	1	5	2	14	25	92	12	42
	Post dinner eating	0	0	1	7	17	63	5	17
	Night eating	0	0	1	7	7	26	3	10
	Sweet eating	0	0	1	7	19	71	12	42
	Social eating	2	10	2	14	6	22	16	57
	Craving for CH	2	10	1	7	21	78	21	75

^§^ Data are expressed as frequencies and percentages. Obesity 1 = Class I patients with obesity; Obesity 2 = Class II/III patients with obesity.

**Table 2 nutrients-11-01995-t002:** Means, standard deviations and quartiles of serum cytokines in relation to BMI.

		IL-2	IL-4	IL-6	IL-8	IL-10	VEGF	INF-γ	TNF-α	IL-1α	IL-1β	MCP1	EGF
Underweight	Mean	5.04	3.03	3.60	14.67	1.51	240.36	0.95	2.70	0.68	1.28	233.58	87.99
	SD	4.62	1.26	3.51	11.60	0.91	144.91	1.51	1.59	0.56	1.72	114.98	108.36
	Q1	0.00	2.47	1.23	8.05	1.42	148.56	0.00	2.00	0.00	0.00	171.03	3.55
	Q2	4.19	2.72	1.97	12.83	1.68	226.40	0.00	2.53	0.80	0.00	220.35	45.57
	Q3	7.41	3.21	4.81	16.85	1.84	331.48	2.25	3.30	1.11	1.69	290.25	147.69
Normal weight	Mean	5.41	2.97	3.01	12.00	2.32	264.96	2.02	3.91	0.86	0.77	346.99	120.76
	SD	7.73	2.65	2.63	8.88	3.13	178.09	3.22	2.45	1.23	1.36	172.86	100.04
	Q1	0.00	0.96	0.96	5.73	1.47	149.38	0.00	2.44	0.00	0.00	211.18	15.72
	Q2	4.56	2.65	2.50	9.63	1.61	191.34	0.00	3.56	0.00	0.00	259.12	123.27
	Q3	8.14	4.08	4.12	16.29	2.48	342.22	3.65	5.38	1.60	1.67	462.60	193.39
Obesity 1	Mean	6.55	4.30	3.41	25.55	2.80	290.74	1.79	2.87	0.69	1.07	297.88	100.43
	SD	9.68	7.67	4.57	52.37	5.38	204.56	2.64	1.65	0.73	1.47	121.83	109.79
	Q1	0.00	1.74	1.58	6.67	0.00	137.89	0.00	2.29	0.00	0.00	209.27	6.18
	Q2	1.95	2.75	2.48	12.01	1.61	272.94	0.00	2.84	0.82	0.00	289.85	61.82
	Q3	9.58	3.68	3.38	19.21	2.90	362.69	2.87	3.93	1.09	2.22	394.14	158.78
Obesity 2	Mean	2.78	1.96	3.78	18.14	1.18	305.44	1.51	2.61	0.42	0.79	282.94	128.66
	SD	3.23	1.56	3.38	31.08	1.21	221.38	2.43	1.58	0.55	1.18	146.39	132.76
	Q1	0.00	1.09	1.16	4.86	0.00	106.09	0.00	1.90	0.00	0.00	173.48	8.76
	Q2	2.69	1.69	2.45	7.51	1.45	291.37	0.00	2.68	0.00	0.00	271.92	81.43
	Q3	4.67	2.67	5.70	14.36	1.88	465.54	2.24	3.79	0.82	1.88	354.55	247.27

Data are expressed as pg/mL. Obesity 1 = Class I patients with obesity; Obesity 2 = Class II/III patients with obesity.

**Table 3 nutrients-11-01995-t003:** Regression analysis.

Dependent Variable	*R* ^2^	F	*p*	Independent Predictors	Beta	*T*	*p*
IL-8	0.077	8.212	0.005	Night eating	0.295	2.886	0.005
IL-10	0.07	4.259	0.007	Post-dinner eating	0.327	2.788	0.007
				Sweet eating	−0.243	−2.07	0.041
IFN-γ	0.034	4.067	0.047	Long fasting	0.213	2.017	0.047
EGF	0.071	7.603	0.007	Night eating	0.285	2.757	0.007

Abbreviations: IL—interleukin; TNF—tumor necrosis factor; IFN-γ—interferon-γ; EGF—Epidermal growth factor.

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
