# Peer review of "Brain-Behavior-Immune Interaction: Serum Cytokines and Growth Factors in Patients with Eating Disorders at Extremes of the Body Mass Index (BMI) Spectrum"

_nutrients, 2019, doi:10.3390/nu11091995_

Round 1
Reviewer 1 Report
The authors describes the relationship between abnormal eating behaviors and cytokines in eating-disordered patient, and it is the strength of this study. The sample size is small and the conclusions are in exploratory stage, but this article will induce the next steps.
The manuscript should be improved in some points listed below.
1) In Introduction, the authors introduce previous reports that IL-6 escalates in AN and IL-6 correlates to BMI, but, these findings contain contradiction. The authors states that such contradiction motivated them to make their hypothesis that eating behaviors might effect to cytokines, however, in Results IL-6 doesn't differ in groups and no picked-up eating behavior explains the IL-6 level. The author should make more precise comments about these mismatches.
2) The Introduction section seems a little redundant (ex. page 2 lines 49-55) and should be improved.
3) Samples were acquired from treatment-seeking subjects. The population may differs from previous studies and this might effect to the results. This point should be stated as limitation.
4) Font size is not unified in many points (ex page 1 lines 24 to 24, page 2 line 47 and line 86...). The abbreviation of "interferon" is not unified (Both "IFN" and "INF" appear.).
5) A redundant space is found in page 5 line 186.
6) The show of decimal point is not unified. Both dot "." and comma "," appear.
6) In page 6 lines 221 to 222 the authors should state that ANBP is also excluded. And, in this sentence, the author should explain the mechanism of the interaction between cytokine levels and purging behaviors, or adequate references which explain this mechanism should be cited.
7) The Discussion section, especially about IL-10 (page 8 lines 289 to 319) is redundant and should be organized.
Author Response
The authors describes the relationship between abnormal eating behaviors and cytokines in eating-disordered patient, and it is the strength of this study. The sample size is small and the conclusions are in exploratory stage, but this article will induce the next steps.
Answer: Thank you
Question 1) In Introduction, the authors introduce previous reports that IL-6 escalates in AN and IL-6 correlates to BMI, but, these findings contain contradiction. The authors states that such contradiction motivated them to make their hypothesis that eating behaviors might effect to cytokines, however, in Results IL-6 doesn't differ in groups and no picked-up eating behavior explains the IL-6 level. The author should make more precise comments about these mismatches.
Answer 1: Thank you for this important comment. We reviewed previous existing results in this field in the introduction, but unfortunately we did not confirm any of these findings in our sample, probably because of the small sample size. So, according to your suggestion, we modified Introduction and Discussion as follows:
Introduction
- Line 55-58: “Previous studies have shown partial links between excess weight gain or body fat mass and increased levels of circulating cytokines, such as IL-6 and TNF-α [12,13]. Only in one study IL-6 levels also correlated positively with body mass index (BMI) [14]”.
-Discussion
-Line 286-288: "Despite previous findings [12,13,14], we did not find any significant differences regarding IL-6 and TNF-a serum levels in our sample, probably because of the small sample size".
Question 2) The Introduction section seems a little redundant (ex. page 2 lines 49-55) and should be improved.
Answer 2: Thank you for this suggestion. We have tried to make it easier and have modified the paragraph as follows:
Line 49-52: “In particular, elevated Tumor Necrosis Factor-alpha (TNF-α) and interleukin (IL)-6 plasma levels and lower levels of IL-2 and Tumor Growth Factor-beta2 (TGF-2) in patients with AN compared to healthy controls (HCs) [5,6,10], probably linked to impaired nutrition and weight loss, [10] have been described.”
Question 3) Samples were acquired from treatment-seeking subjects. The population may differ from previous studies and this might effect to the results. This point should be stated as limitation.
Answer 3: Thanks for this comment, although we do not completely agree with the reviewer, we have added this point in " Strengths and Limitations" at line 344-345: "Furthermore, we have to specify that our sample was acquired from treatment-seeking subjects, so this population may differ from previous studies and this might effect to results. "
Question 4) Font size is not unified in many points (ex page 1 lines 24 to 24, page 2 line 47 and line 86...). The abbreviation of "interferon" is not unified (Both "IFN" and "INF" appear.).
Answer 4: Thanks for these suggestions. We have corrected the text as specified.
Question 5) A redundant space is found in page 5 line 186.
Answer 5: We thank the reviewer for this correction. We have modified text as suggested.
Question 6) The show of decimal point is not unified. Both dot "." and comma "," appear.
Answer 6: We have corrected all decimal point indicated with "dots" into "commas".
Question 7) In page 6 lines 221 to 222 the authors should state that ANBP is also excluded. And, in this sentence, the author should explain the mechanism of the interaction between cytokine levels and purging behaviors, or adequate references which explain this mechanism should be cited.
Answer 7: Thanks for this suggestion. Unfortunately, to our knowledge there are not studies that evaluated serum cytokines levels specifically in purging AN or BN, but medical complication of these dysfunctional behaviours are well known. We have clarified our discussion as follows:
Line 215-220: "We excluded patients with BN because levels of cytokines could be influenced by other medical comorbidities linked to frequent long term purging behaviors, self-induced vomiting and laxatives or diuretics use, or prolonged periods of starvation that could influence inflammatory response and immune system, causing chronic inflammation of oral and digestive system and impairment of liver and kidney function [28, 29]".
We have also added two of the most recent studies on this topic in the reference list that could help the reader.
Castillo, M.; Weiselberg, E. Bulimia Nervosa/Purging Disorder. Curr Probl Pediatr
Adolesc Health Care2017, 47, 85-94. Westmoreland, P.; Krantz, M.J.; Mehler, P.S. Medical Complications of Anorexia
Nervosa and Bulimia. Am J Med2016, 129, 30-7.
Question 8) The Discussion section, especially about IL-10 (page 8 lines 289 to 319) is redundant and should be organized.
Answer 8: We reassumed the paragraph as follows:
Line 293-312: "IL-10 synthesized by several cell types within multiple organs including the spleen, is a potent anti-inflammatory cytokine [53]. Lower IL-10 production capacity has been demonstrated in obesity [16], implicating that the lack of IL-10 may, at least in part, contribute to the prevalent pro-inflammatory profile expressed in obesity [54,55].
We found lower IL-10 levels in obese subjects (with and without BED) compared to AN and HCs, confirming previous findings [55]. On the other hand, elevated IL-10 levels have been reported in many inflammatory conditions, including malnutrition [18], and may indicate attempts to inhibit inflammation by downregulating the secretion of pro-inflammatory cytokines [18].
Few previous studies have evaluated IL-10 in AN patients [9]. We found higher levels of IL-10 in AN patients compared to obese and HCs. We hypothesize that our result, not yet clarified by previous studies, may be a consequence of malnutrition.
In fact, the overproduction of IL-10 and the downregulation of pro-inflammatory cytokines have often been associated with immunosuppression [54-58]. IL-10 is a particularly important anti-inflammatory mediator that regulates both innate and adaptive immune responses [57,59-61] with a severe reduction in spleen cellularity [59]. Previous studies found different patterns of immune markers variability among obese patients, during progressive weight loss [27,61].
Finally, we evaluated possible predictors of higher cytokine levels. The association with IFN-γ and long term fasting have been underlined before by us, and confirmed by a previous study [7]. We hypothesize that elevated levels of serum IL-10 in not habitual sweet eaters, could be inversely proportional to metabolic and weight consequences of this eating behavior. "
Finally, we corrected the title as underlined in Yellow, because of mistakes in spaces (lines 1-2).

Reviewer 2 Report
This manuscript "Brain-behavior-immune interaction: serumcytokines and growth factors inpatients with eating disorders at extremes of the body mass index (BMI) spectrum" is a good study with some novel findings that is important to the field.
My comments
1) It would be very important to sort the data according to gender as several of the parameters tested in this study can vary significantly between males and females.
2) Also the range of age chosen for this study is very wide, some of these parameters would also vary with aged so dividing the data into age groups e.g. 18-40 and 40-65 then re-analysis accordingly would give better interpretation to the data.
Author Response
This manuscript "Brain-behavior-immune interaction: serumcytokines and growth factors inpatients with eating disorders at extremes of the body mass index (BMI) spectrum" is a good study with some novel findings that is important to the field.
Answer: Thank you very much for this positive comment.
Question 1) It would be very important to sort the data according to gender as several of the parameters tested in this study can vary significantly between males and females.
Answer: we fully agree with the reviewer. We did not make a further subdivision in the tables according to sex because of the small sample size . Anyway, we controlled for this variable as a fixed factor in the Anova and no significant differences emerged regarding any cytokine.
Question 2) Also the range of age chosen for this study is very wide, some of these parameters would also vary with aged so dividing the data into age groups e.g. 18-40 and 40-65 then re-analysis accordingly would give better interpretation to the data.
Answer: Again, thank you very much for this suggestion: we do agree with the reviewer that age is an important variable so we have controlled for age as a covariate in the Anova. We have considered age as a covariate because splitting the variable into 2 ranges, as suggested, could create some errors in the analysis as patients with AN and BED are usually, but not always, at the extremes. Anyway, the results did not demonstrate any significant difference for any cytokine. In the text we have modified the new values of F, p and Ćž2 after controlling for sex and age. Somehow, the small sample size is a limit that we have highlighted.

Round 2
Reviewer 1 Report
The authors have responded clearly to my previous review.
This version of manuscript is worthy to be published after correcting very minor points.
Redundant spaces (ex. line 188 and line 196) should be fixed.
Font size should be unified (lines 58 - 59).
Author Response
The authors are grateful for the reviewer's positive comments. Thanks!